# Transplant and Recipient Factors in Prediction of Kidney Transplant Outcomes: A UK-Wide Paired Analysis

**DOI:** 10.3390/jcm11082222

**Published:** 2022-04-15

**Authors:** Richard Dumbill, Roderick Jaques, Matthew Robb, Rachel Johnson, Rutger J. Ploeg, Maria E. Kaisar, Edward J. Sharples

**Affiliations:** 1Nuffield Department of Surgical Sciences, University of Oxford, Oxford OX3 9DU, UK; rutger.ploeg@nds.ox.ac.uk (R.J.P.); maria.kaisar@nds.ox.ac.uk (M.E.K.); 2Oxford Transplant Unit, Oxford University Hospitals, Oxford OX3 7LE, UK; edward.sharples@ouh.nhs.uk; 3Statistics and Clinical Studies, NHS Blood and Transplant, Bristol BS34 7QH, UK; roderick.jaques@nhsbt.nhs.uk (R.J.); matthew.robb@nhsbt.nhs.uk (M.R.); rachel.johnson2@nhsbt.nhs.uk (R.J.); 4Research and Development, NHS Blood and Transplant, Oxford OX3 9DU, UK

**Keywords:** kidney transplantation, graft function, paired, delayed graft function

## Abstract

Background: In kidney transplantation, the relative contribution of various donor, procedure and recipient-related factors on clinical outcomes is unknown. Previous paired studies have largely focused on examining factors predicting early outcomes, where the effect of donor factors is thought to be most important. Here, we sought to examine the relationship between early and long-term outcomes in a UK-wide paired kidney analysis. Methods: UK Transplant Registry data covering 24,090 kidney transplants performed between 2001–2018, where both kidneys from each donor were transplanted, were analysed. Case-control studies were constructed using matched pairs of kidneys from the same donor discordant for outcome, to delineate the impact of transplant and recipient factors on longer-term outcomes. Results: Multivariable conditional logistic regression identified HLA mismatch as an important predictor of prolonged delayed graft function (DGF), in the context of a paired study controlling for the influence of donor factors, even when adjusting for early acute rejection. Prolonged DGF, but not human leucocyte antigen (HLA) mismatch, strongly predicted 12-month graft function, and impaired 12-month graft function was associated with an increased risk of graft failure. Conclusions: This study indicates prolonged DGF is associated with adverse long-term outcomes and suggests that alloimmunity may contribute to prolonged DGF by a mechanism distinct from typical early acute rejection.

## 1. Introduction

Early post-transplant events are important determinants of graft survival and function [1]. Ischaemia-reperfusion injury (IRI) is a universal insult in deceased donor kidney transplantation, and initially manifests by slow improvement in function, or, when more severe, delayed graft function (DGF), normally defined by the need for dialysis therapy in the first seven days. Ischaemia-reperfusion may injure the graft by various mechanisms leading to tubular cell death, including innate immune activation [2]. This early inflammatory response promotes the development of adaptive immunity and acute rejection, which predispose the graft to progression of interstitial fibrosis and tubular atrophy [3]. Demographic change has led to an increase in donor age, comorbidity and risk index, and there is a trend towards higher utilisation of organs from donors deceased after circulatory death [4]. These factors mean that the clinical manifestations of IRI may be increasing. Donor factors play an important role in the early status of the transplanted kidney, and these early events have long-term sequelae in organ function, risk of later adverse events, and ultimately graft survival [5]. Recipient factors and differences among care protocols after transplant also dictate outcome [6]; however, the relative contributions for the risk of graft failure derived from the donor kidney versus the new immunologic milieu of the recipient (or other non-donor effects) are unclear.

Outcome data concerning transplantation of pairs of kidneys donated by a common deceased donor to different recipients enables assessment of the relative contribution of donor and non-donor factors. Paired kidney analysis has therefore been used to examine specific predictors that are discordant between recipients [7,8], outcomes that are discordant between recipients [9], or the relative contribution from donor and recipient factors towards an outcome [10,11,12]. If the concordance for pairs of kidneys is low, close to the level expected by chance, then this would suggest a minimal effect of donor characteristics, and that non-donor factors are superior. Small studies, often from single centres, have used pairs of transplanted kidneys to quantify donor effects, and have focused on outcomes such as delayed graft function. Traynor et al. examined the correlation between 652 pairs of donor kidneys for estimated glomerular filtration rate (eGFR) at 12 months, with similar Spearman’s co-efficient at 12 and 60 months, suggesting a significant relationship in transplant function within pairs [13]. In larger registry studies, OPTN and USRDS data showed that for some early outcomes, such as delayed graft function, kidney pairs are likely to show concordant outcomes, with the second kidney having between 1.7–2.05 relative risk of DGF if the other kidney had developed DGF [10,11]. One- and three-year graft survival showed only a minor excess risk, indicating that other recipient factors, peri-operative differences, or events after transplantation were more likely to have a causal link to outcome, suggesting that unmeasured donor characteristics contribute significantly to allograft failure.

DGF is well known to occur at different frequencies, and carry different significance, in donation after brain death (DBD) and donation after circulatory death (DCD) donor kidneys [14,15]. It is increasingly recognised that the clinical syndrome of DGF represents a number of different clinicopathological entities and causes, which likely explains this heterogeneity [16]. However, irrespective of mechanism, prolonged DGF represents more substantial damage than brief DGF, which is reflected in the findings reported by Philips et al. who showed that, even in the DCD cohort, DGF of more than two weeks is an adverse prognostic indicator portending worse outcomes than short or no DGF [17]. Paired analysis can be used, in the case of a dichotomous outcome observed differentially within pairs of donor kidneys, to construct natural experiments controlling for donor factors, even when these are unmeasured, more precisely delineating the contribution from transplant and recipient factors. In this study, we aimed to use these methods to examine transplant and recipient factors that are associated with measures of graft function—prolonged DGF and 12-month eGFR—that are known to predict longer-term outcomes.

## 2. Materials and Methods

The study used data supplied by the NHSBT Transplant Registry. All analyses were performed using a fully anonymised extract with no identifiable information available to the research team; therefore, institutional review board approval was not required. The registry includes data on all donors and transplant recipients in the UK. The initial study population consisted of all UK deceased-donor renal transplants performed between 1 January 2001 and 1 January 2018 where both donor kidneys were transplanted. All transplantation centres in the United Kingdom were represented and therefore geographical variations were minimized. For the primary analysis (prolonged DGF), exclusions were made where records had missing DGF data, primary non-function (PNF) or graft survival was less than 30 days, or where recipients were pre-dialysis at the point of transplantation. The dataset was then reduced to consider only those transplants where both kidneys from each donor met these inclusion criteria. Finally, a DGF-discordant dataset was created by extracting those cases where outcomes were discordant within pairs—in this case, where one donor kidney from each pair went on having prolonged DGF, defined as greater than or equal to 14 days, and the other donor kidney had DGF of less than 14 days or no DGF.

An equivalent dataset was constructed for analysis of 12-month eGFR by defining good graft function as an eGFR of greater than or equal to 45 mL/min/1.73 m^2^, and poor graft function as less than 45 mL/min/1.73 m^2^. Exclusions were made for missing 12-month eGFR data, or graft survival under 1 year. Again, transplants were only included if both donor kidneys met these inclusion criteria. A 12-month function-discordant dataset was then generated, where, within each transplant pair, one transplant had resulted in a recipient 12-month eGFR of greater than 45 mL/min/1.72 m^2^, and the other in a recipient 12-month eGFR of less than 45 mL/min/1.73 m^2^.

DGF was defined as the need for dialysis within seven days of transplantation. Duration was defined as the number of days between transplantation and the last dialysis treatment where recipients subsequently attained dialysis independence. Prolonged DGF was defined as that lasting ≥ 14 days. Rejection was defined as a treated, clinically diagnosed rejection episode. Recipients were defined as being highly sensitised if they had a calculated reaction frequency > 85%. eGFR was recalculated from raw data recorded in the registry using the CKD-EPI formula [18]. HLA mismatch grades were used as defined by NHSBT [19] (see Appendix A).

### Statistical Analysis

R version 4.0.2 was used for the statistical analysis. The raw variables were cleaned. Univariate analyses were performed where appropriate, comparing categorical variables using chi-square tests, and continuous variables with *t*-tests. Where data was missing, variables were inspected for the pattern of missingness, and the assumption that data was missing at random was tested. Missing data was dealt with by means of multiple imputation by chained equations, using the MICE package for R with m = 10; this procedure was applied after initial inclusion criteria were applied for each study, but before any records with intact outcome variables were excluded based on a missing contralateral kidney, or due to outcome concordance within a pair. This approach maximises the amount of information available for use during imputation. Following an initial imputation round, model convergence was inspected, and the predictor subset and imputation methods refined as necessary as described elsewhere [20]. The multiply imputed datasets were then restricted to discordant pairs as described above. Conditional logistic regression models were constructed using the outcome variable of interest as the independent variable, clinically relevant parameters as predictors, and with strata applied at the level of the donor.

## 3. Results

### 3.1. Missing Data

The pattern of missing data for clinically important predictors was examined (see Appendix A). In general, donor data was well-recorded, with the fraction of missing information ranging from 0% for donor age and sex to 8.8% for donor retrieval creatinine. We noted that there was a significant difference for several donor variables (donor BMI, donor retrieval creatinine, donor history of hypertension, donor history of diabetes, donor history of smoking) between the probability of observing the primary adverse outcome of interest (prolonged DGF) in those patients where the variable was observed versus those where the data was missing. In each case, the frequency of prolonged DGF was higher where data was missing. Pre-transplant recipient information was very well recorded with recipient BMI being the only variable with a substantial amount of missing information (39.2%). The frequency of prolonged DGF was not different where recipient BMI was observed versus missing. Use of a paired-analysis strategy provides benefit where donor factors are not missing at random or are unmeasured.

### 3.2. DBD and DCD Cohorts

We performed an exploratory analysis of the DBD and DCD cohorts separately, to test the hypothesis that, when considering prolonged DGF defined using a cut-off of 14 days as the outcome of interest, it is reasonable to combine these cohorts, in line with the findings of Philips et al. [17]. For univariate analysis, prolonged DGF was associated in both DBD and DCD donors with reduced 12-month eGFR (DBD: 51.0 vs. 39.8, *p* < 0.001; DCD: 49.6 vs. 40.3, *p* < 0.001). Prolonged DGF was also associated with impaired long-term graft survival in both the DBD and DCD cohorts (Appendix A). In a multivariate survival analysis, donor type was weakly associated with hazard of graft failure when the model included only donor type and prolonged DGF terms (HR 0.91, *p* = 0.02); however, when donor and recipient age were included, there was no residual independent effect of donor type (HR 0.96, *p* = 0.42). Prolonged DGF remained an important predictor (HR 2.12, *p* < 0.001). We therefore concluded that prolonged DGF carried sufficiently similar implications in the DBD and DCD cohorts for them to be combined in our paired analysis, which, by design, controls for donor type.

### 3.3. Prolonged DGF Discordant Pairs

#### 3.3.1. Cohort

There were 24,090 kidney transplants performed in the United Kingdom between 1 January 2001 and 1 January 2018 where both donor kidneys were transplanted. Demographic information for the entire cohort is presented in Appendix A. Those transplants resulting in PNF, with graft survival < 30 days, in recipients who were pre-dialysis at transplant, and where DGF data were missing, were excluded. After removing those cases which no longer had a partner kidney in the dataset, 12,668 cases remained. For the primary analysis, the subset of cases where donor pairs were discordant for prolonged DGF were selected; this amounted to 1328 transplants suitable for inclusion in a paired case-control study. This subset selection process is illustrated in Figure 1.

#### 3.3.2. Univariate Analysis

The donor demographics for this study population are shown in Table 1. The mean age of donors was 52.5 years. They were predominantly male (62.3%), and 48.3% were DCD donors. The mean age of recipients was 52.1, 64.3% were male, and they had an average wait-time of 1108 days. Results of a univariate analysis examining association between transplant and recipient factors with prolonged DGF in this cohort are shown in Table 1. By design, 664/1328 (50%) recipients experienced prolonged DGF (≥14 days). A total of 265 of the paired transplants (20%) resulted in short-duration DGF, whilst 399 (30%) had immediate graft function.

#### 3.3.3. Multivariate Analysis

The results of a multiply imputed multivariate analysis considering this paired discordant cohort are shown in Table 2. Longer cold ischaemia time, recipient history of diabetes, longer wait-list time, previous renal transplant, recipient dialysis modality being HD, not using tacrolimus, and higher categories of HLA mismatch were all independently associated with prolonged DGF. Inclusion of early acute rejection (within the first 14 days) in the model showed that the effect of HLA mismatch on prolonged DGF was independent of early diagnosis of rejection. The significant effect of poorer HLA mismatch on prolonged DGF persisted when HLA mismatch group was substituted for total number of HLA mismatches (Appendix A), suggesting that the small group size of some of the HLA mismatch groups was not responsible for the effect.

A dichotomous outcome is necessary for paired analysis of this sort, so DGF cannot be treated as a three-level categorical variable (primary function vs. short-duration DGF vs. long-duration DGF). This paired analysis mandates choice of a cut-off value, and the comparator group (short or no DGF) is necessarily heterogenous. We therefore repeated the analysis considering DGF duration as a binary variable ‘longer’ or ‘shorter’ within each pair, thereby avoiding a threshold effect (Appendix A). Primary function was treated as a DGF duration of 0 days. Kidney donor pairs with DGF duration information missing were excluded, as were pairs where DGF duration was equal (or both were primary function). A total of 6682 transplants were included. Again, this analysis showed an associated between greater HLA mismatch and longer DGF, independent of early acute rejection.

### 3.4. Graft Function at 12 Months

#### 3.4.1. Cohort

The mean 12-month eGFR in those patients in analysis 1 who experienced prolonged DGF was 41.2 mL/min/1.73 m, compared to 46.2 mL/min/1.73 m^2^ for those who did not (*p* < 0.01 for univariate analysis). We therefore elected to dichotomise 12-month graft function at the KDIGO CKD 3a/3b cut-off of 45 mL/min/1.73 m^2^ for the purposes of generating a case-control cohort determined on good or poor graft function. Good 12-month function was defined as an eGFR of greater than or equal to 45 mL/min/1.73 m^2^, poor function as less than 45 mL/min/1.73 m^2^.

A total of 5276 transplants were identified for inclusion in this case-control study after excluding those with missing 12-month eGFR data, those with graft survival of less than one year, those missing a paired donor kidney, and those with concordant 12-month graft function (Figure 2).

Figure 2 represents a consort-type flow diagram showing selection of the case-control cohort for analysis 2 (good vs. poor 12-month function, defined using an eGFR cut-off of 45 mL/min). A total of 5276 outcome-discordant transplants were identified for inclusion in this study.

#### 3.4.2. Univariate Analysis

Those recipients with good 12-month graft function were more likely to be male, with a lower BMI, to have received tacrolimus and prednisolone at transplant, and to have received a right donor kidney. They were less likely to be recipients of a retransplant, to be highly sensitised, or to have experienced prolonged DGF. The donors included in this study cohort were an average age of 51.7, were slightly more likely to be male (51.9%), and 31.3% were DCD donors. These population characteristics and between-group differences are shown in Table 3.

#### 3.4.3. Multivariate and Graft Survival Analysis

The results of a multivariable paired conditional logistic regression analysis examining predictors of 12-month functional category are shown in Table 4. Prolonged DGF, female recipient sex, an early rejection episode, receipt of a left donor kidney, being retransplanted, and higher recipient BMI were all independent predictors of poor 12-month graft function. Tacrolimus and prednisolone use at transplant were protective. Of note, the HLA mismatch group did not predict 12-month functional category. A total of 286 subsequent graft failures were observed in the cohort with good 12-month function, compared to 550 graft failures in the poor 12-month function group. There was a highly significant difference in long-term graft survival between these two groups (log-rank test *p* < 0.0001); Figure 3.

## 4. Discussion

This study aimed to estimate the impact of transplant and recipient factors on graft function and outcomes, using paired kidneys from a large national registry, in the modern era when acceptance of older and higher risk donor kidneys in deceased donor transplantation is increasing. Where the outcome of interest is binary, observation of outcome-discordant donor kidney pairs allows construction of a natural experiment which controls for donor factors. This is important for control of unmeasured donor factors but also for dealing with missing data; our analysis shows that the pattern of missingness with respect to donor variables is not random, and, in fact, missingness is associated with observation of adverse outcomes of interest. Use of a study design that abrogates the impact of missing donor factors, as well as controls for those which are unmeasured, appears therefore to be highly desirable.

Prolonged DGF has previously been shown to be an important predictor of longer-term outcomes, even in the DCD cohort [17]. This is biologically plausible, regardless of the cause of the clinical syndrome of DGF, as the magnitude of injury should be expected to be related to its duration. Choice of prolonged DGF as an outcome measure therefore minimises any interaction between donor type and significance of the endpoint. It is notable that a key predictor of prolonged DGF appears to be HLA mismatch, with adverse matches predicting a higher chance of prolonged DGF. Further, it appears that this effect is independent of diagnosis of conventional early acute rejection. Innate inflammation and complement activation are known to be contributors to the development of DGF [21,22,23]. It is possible that this relationship with HLA mismatch represents an adaptive immune component to DGF, causing cellular injury and contributing to its persistence beyond two weeks, which is a different entity to typical acute rejection. This model is consistent with previous reports demonstrating T-cell infiltration and subclinical rejection in protocol biopsies from patients with DGF, which, in combination, are associated with interstitial fibrosis [24]. There is also known to be crosstalk between alloantigen presentation and complement activation in DGF, again illustrating a potential mechanism [25].

The effect of prolonged delayed graft function on 12-month graft function was restated in our paired analysis. Further, in this paired case-control cohort without donor-derived effects, 12-month function was highly significant for prediction of long-term graft survival. Our interpretation is that prolonged DGF is a good surrogate for cumulative peri-transplant graft injury which is in part due to an adaptive recipient immune effect distinct from typical rejection. The effect of early graft injury sufficient to cause DGF of more than two weeks is detectable and important when considering 12-month graft function, which, when impaired, predicts early transplant failure.

Recipient and transplant variates other than HLA mismatch which predicted prolonged DGF are generally associated with a pro-inflammatory recipient state (recipient diabetes, longer wait-list time, previous renal transplant), consistent with previous reports [26]. The influence of recipient dialysis modality on probability of developing prolonged DGF is interesting but difficult to interpret. A relationship between probability of DGF and pre-transplant dialysis modality has previously been observed [27,28] and is supported by this study. The possibilities include that this is due to better preservation of native renal function in peritoneal dialysis (PD) patients, that it is due to a pro-inflammatory effect of HD, or, simply, that reporting of PD use post-transplant, which by its nature is much more frequently self-administered than haemodialysis (HD), is more likely to be missed in this registry dataset. The finding that 12-month eGFR is predicted by recipient sex and donor kidney side is unexpected. It is possible that the effect of recipient sex is mediated by the equation used to estimate GFR from serum creatinine, which includes sex as a predictor [18]. It is also possible that the effect of donor kidney side is related to a tendency for technically more challenging recipients to be transplanted with a left donor kidney (with a longer vein and shorter artery than in the right kidney) where there is a choice, due to preserved vessel length for implantation. There is known to be a small increased risk of DGF, and early graft loss, associated with right kidneys [29]. We observed that mean recipient BMI in recipients of a left kidney was higher than recipients of a right kidney (26.3 vs. 26.9; *p* < 0.001), which may reflect a tendency to transplant right kidneys into technically easier recipients where possible; this may translate to better 12-month function.

There are a number of limitations of this study. Firstly, we were unable to include the effect of induction immunosuppression or anastomotic time due to a high proportion of missing data and inconsistent coding in the registry. Secondly, some variables (particularly recipient BMI) required significant imputation due to missing data. Despite these limitations, our data shows the residual importance of transplant and recipient variables on prediction of meaningful long-term outcomes in kidney transplantation, and that HLA matching and alloreaction may be an important mediator of outcomes, much earlier than previously recognised, by a mechanism distinct from typical acute rejection.

## Figures and Tables

**Figure 1 jcm-11-02222-f001:**
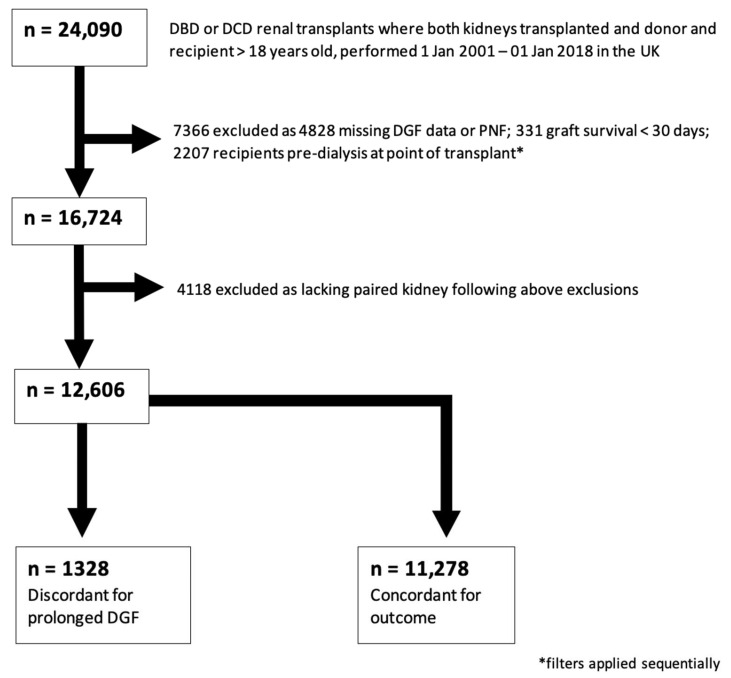
**Study Population Selection (prolonged DGF).** Consort-type flow diagram showing selection of the case-control cohort for analysis 1 (prolonged DGF). 1328 outcome-discordant transplants were identified for inclusion in this study.

**Figure 2 jcm-11-02222-f002:**
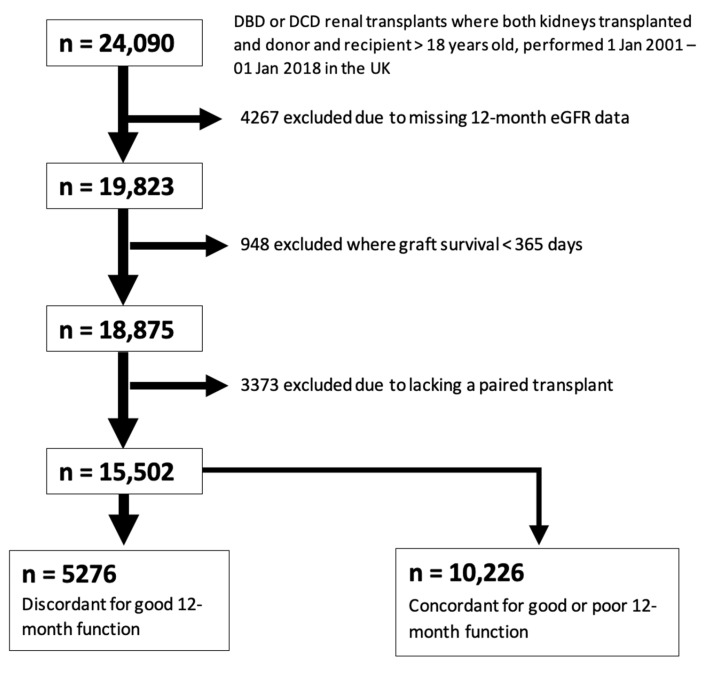
Study Population Selection (12-month eGFR).

**Figure 3 jcm-11-02222-f003:**
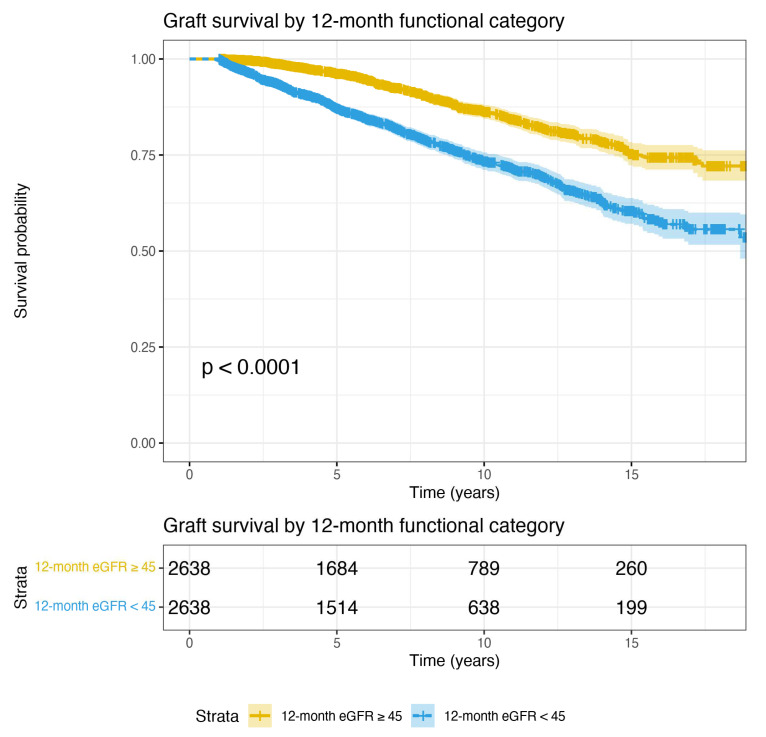
Graft survival curves for the analysis 2 cohort (discordant on 12-month function), up to 18 years post-transplant. 12-month survival is 100% in this case-control study cohort by selection. Again, by design, one kidney from each donor is included in each of the two groups. This figure shows that when donor factors are controlled, 12-month functional category defined using an eGFR cut-off of 45 mL/min is a highly significant predictor of graft survival (*p* < 0.0001, log-rank test).

**Table 1 jcm-11-02222-t001:** Univariate analysis—Primary Study Cohort (pairs discordant for prolonged DGF).

Parameter	DGF < 14 Days or Primary Function	Prolonged DGF	*p*-Value for the Difference
*N*	664	664	-
*Donor age, years, mean (sd)*	52.5 (13.8)	52.5 (13.8)	-
*Donor sex, male, n (%)*	414 (62.3)	414 (62.3)	-
*Donor BMI, mean (sd)*	27.2 (5.0)	27.2 (5.0)	-
*Donor type, DCD, n (%)*	321 (48.3)	321 (48.3)	-
*Donor creatinine at retrieval*	93.1 (69.7)	93.1 (69.7)	-
*Donor side, right, n (%)*	334 (50.3)	330 (49.7)	0.87
*HLA mismatch group, n (%)*			0.001
*Group 1*	57 (8.6)	39 (5.9)	
*Group 2*	249 (37.5)	198 (29.8)	
*Group 3*	297 (44.7)	361 (54.4)	
*Group 4*	61 (9.2)	66 (9.9)	
*Cold ischaemic time, hours, mean (sd)*	15.8 (5.3)	17.2 (5.9)	<0.001
*Recipient age, years, mean (sd)*	52.1 (13.1)	52.2 (13.0)	0.90
*Recipient sex, male, n (%)*	420 (63.3)	434 (65.4)	0.46
*Recipient BMI, mean (sd)*	26.6 (4.7)	27.4 (4.9)	0.01
*Recipient dialysis modality PD at transplant, n (%)*	205 (30.9)	123 (18.5)	<0.001
*Recipient diabetes, n (%)*	52 (7.8)	79 (11.9)	0.02
*Recipient wait-time, years, mean (sd)*	2.7 (2.3)	3.3 (2.8)	<0.001
*Retransplant, YES, n (%)*	73 (11.0)	128 (19.3)	<0.001
*Highly sensitised recipient, n (%)*	45 (6.8)	66 (10)	0.045
*HLA incompatible transplant, n (%)*	3 (0.5)	18 (2.7)	0.002
*Tacrolimus at transplant, n (%)*	546 (82.4)	523 (78.9)	0.13
*Prednisolone at transplant, n (%)*	558 (84.0)	551 (83.1)	0.70

**Table 2 jcm-11-02222-t002:** Multivariate analysis—Odds of prolonged DGF (mismatch groups).

*Parameter*	OR (95% CI)	Significance (*p*-Value)
*Mismatch group 1 (reference)*	-	-
*Mismatch group 2*	2.59 (1.16–5.78)	0.02
*Mismatch group 3*	3.3 (1.45–7.52)	<0.01
*Mismatch group 4*	2.92 (1.11–7.69)	0.01
*Cold Ischaemia Time (hours)*	1.06 (1.04–1.09)	<0.01
*Waitlist time (years)*	1.12 (1.04–1.21)	<0.01
*Recipient age (years)*	1.01 (0.99–1.02)	0.26
*Recipient sex (male)*	1.18 (0.91–1.54)	0.21
*Recipient BMI (units)*	1.02 (1.00–1.05)	0.06
*Recipient dialysis type (PD)*	0.53 (0.39–0.72)	<0.01
*Recipient diabetes*	1.68 (1.09–2.59)	0.02
*Recipient previous renal transplant*	2.40 (1.53–3.78)	<0.01
*Recipient rejection within 14 days*	2.47 (1.65–3.69)	<0.01
*Recipient highly sensitised*	1.30 (0.70–2.42)	0.40
*Donor organ side (right)*	1.08 (0.90–1.29)	0.42
*HLA incompatible transplant*	3.26 (0.88–12.06)	0.08
*Tacrolimus at transplant*	0.56 (0.36–0.88)	0.01
*Prednisolone at transplant*	0.76 (0.49–1.19)	0.23

**Table 3 jcm-11-02222-t003:** Univariate analysis—Pairs discordant for good 12-month function (cut point 45 mL/min/1.73 m^2^).

	12-Month eGFR > 45 mL/min/1.73 m^2^	12-Month eGFR < 45 mL/min/1.73 m^2^	*p*-Value for the Difference
*n*	2595	2595	- ^1^
*Donor age, years, mean (sd)*	51.68 (12.87)	51.68 (12.87)	- ^1^
*Donor sex, male, n (%)*	1345 (51.8)	1345 (51.8)	- ^1^
*Donor BMI, mean (sd)*	26.90 (5.27)	26.90 (5.27)	- ^1^
*Donor type, DCD, n (%)*	815 (31.4)	825 (31.3)	- ^1^
*Donor creatinine at retrieval*	85.06 (47.81)	85.06 (47.81)	- ^1^
*HLA mismatch group, n (%)*			0.968 ^2^
*Group 1*	339 (13.1)	339 (13.1)	
*Group 2*	973 (37.5)	959 (37.0)	
*Group 3*	1098 (42.3)	1115 (43.0)	
*Group 4*	185 (7.1)	182 (7.0)	
*Cold ischaemic time, hours, mean (sd)*	16.28 (5.82)	16.55 (5.76)	0.099 ^3^
*Recipient age, years, mean (sd)*	50.60 (13.19)	50.90 (12.29)	0.404 ^3^
*Recipient sex, male, n (%)*	1725 (66.5)	1492 (57.5)	<0.001 ^2^
*Recipient BMI, mean (sd)*	25.98 (4.56)	26.98 (4.88)	<0.001 ^3^
*Recipient dialysis modality at transplant, n (%)*			0.538 ^2^
*HD*	1653 (64.0)	1617 (62.6)	
*PD*	658 (25.5)	677 (26.2)	
*Predialysis*	273 (10.6)	291 (11.3)	
*Recipient diabetes, n (%)*	222 (8.6)	203 (7.8)	0.362 ^2^
*Recipient wait-time, years, mean (sd)*	2.67 (2.31)	2.74 (2.48)	0.287 ^3^
*Retransplant, YES, n (%)*	337 (13.0)	393 (15.1)	0.028 ^2^
*Highly sensitised recipient, n (%)*	195 (7.6)	239 (9.3)	0.033 ^2^
*Donor kidney side, right, n (%)*	1360 (52.4)	1235 (47.6)	0.001 ^2^
*HLA incompatible transplant, n (%)*	15 (0.6)	27 (1.0)	0.088 ^2^
*Prolonged DGF, n (%)*	85 (3.9)	159 (7.2)	<0.001 ^2^
*Tacrolimus at transplant, n (%)*	2045 (79.0)	1932 (74.6)	<0.001 ^2^
*Prednisolone at transplant, n (%)*	2229 (86.1)	2168 (83.7)	0.018 ^2^

**Univariate analysis for cohort of transplants discordant for 12-month function.**^1^ Donor factors were not compared between groups, as they are by design identical. ^2^ Categorical variables were compared using Chi-square tests. ^3^ Continuous variables were compared using independent *t*-tests.

**Table 4 jcm-11-02222-t004:** Multivariate analysis—Odds of poor 12-month function.

*Parameter*	Odds Ratio (95% CI)	*p*-Value for the Difference
*Prolonged DGF*	2.03 (1.49–2.78)	<0.01
*HLA mismatch group 1 (reference)*	-	-
*Group 2*	1.05 (0.80–1.39)	0.71
*Group 3*	1.04 (0.78–1.39)	0.80
*Group 4*	1.01 (0.69–1.49)	0.95
*Cold ischaemia time (hours)*	1.01 (1.00–1.02)	0.11
*Waitlist time (years)*	1.00 (0.97–1.04)	0.77
*Recipient age (years)*	1 (1.00–1.01)	0.28
*Recipient sex (male)*	0.66 (0.58–0.74)	<0.01
*Recipient BMI (units)*	1.02 (1.00–1.05)	0.03
*Recipient dialysis type, HD (reference)*	-	-
*PD*	1.05 (0.91–1.20)	0.52
*Predialysis*	1.19 (0.97–1.44)	0.09
*Recipient diabetes*	0.88 (0.71–1.09)	0.25
*Recipient—retransplant*	1.29 (1.07–1.56)	0.01
*Recipient—highly sensitised*	1.12 (0.86–1.46)	0.40
*Donor kidney right (reference left)*	0.91 (0.84–0.99)	0.03
*HLA incompatible transplant*	1.42 (0.74–2.73)	0.29
*Rejection episode within 14 days*	1.66 (1.31–2.11)	<0.01
*Tacrolimus use at transplant*	0.6 (0.5–0.73)	<0.01
*Prednisolone use at transplant*	0.72 (0.59–0.88)	<0.01

## Data Availability

The data used in this study are available from the NHSBT statistics service, subject to application and approval.

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
