# Peer review of "Transplant and Recipient Factors in Prediction of Kidney Transplant Outcomes: A UK-Wide Paired Analysis"

_jcm, 2022, doi:10.3390/jcm11082222_

Round 1
Reviewer 1 Report
Please write what is DBD and DCD. Please write explanation of abbreviation in each Figures.
How about ABO match and apheresis before kidney transplantation.
Immunosuppressive drug is the most key factor in prediction of kidney transplantation. Authors could not include and write this at limitation, however it is need to include.
Author Response
Dear Reviewer,
Many thanks indeed for your review and comments on our manuscript. To address your points in turn:
- Please accept my apologies for failing to include the expansion of abbreviations used in the manuscript; I had overlooked this when completing the manuscript template. In the revised version of the manuscript I have included a table of abbreviations.
- ABO mismatch is extremely uncommon in the UK in the setting of deceased-donor kidney transplantation, as there is insufficient time pre-operatively for the necessary apheresis procedure to be performed. I interrogated our dataset and found only three recorded instances of ABOi transplants, out of our dataset which consists of 23,700 cases. Therefore, there is not enough data to include this as a variable in the models.
- As for immunosuppression - I agree that this is important. I have been fortunate enough to obtain some relevant data on this in our study cohort. I have updated the manuscript to include data and results on immunosuppression at the time of transplant. This has not changed our conclusions - although as you suggest, this is an additional factor which is related to outcomes (prolonged DGF and 12-month function)
Reviewer 2 Report
The publication "Transplant and recipient factors in the prediction of kidney transplant outcomes: A UK wide paired analysis" presents the important issue of finding predictors between early and late outcomes in paired kidney analysis. Although similar studies have already been published, the work reviewed primarily examines the influence of various factors on the late effects of kidney transplant failure. Noteworthy is a thorough statistical analysis and the use of many statistical models. However, there is a lack of, for example, a graphic representation of the ROC curve analysis showing the usefulness of prediction in a graphical manner. Moreover, a limitation of the study is the lack of taking into account the course of immunosuppressive pharmacotherapy, dose adjustment based on the results of monitoring drug concentrations in the recipient's organism and, possibly, the influence of interactions between immunosuppressants and other factors. Overall, the results encourage a further search for biomarkers that predict delayed graft function. The text requires stylistic and punctuation refinement.
Author Response
Dear Reviewer,
Many thanks for your comments. We have not included ROC analysis as the objective of the manuscript was not outcome prediction - it was to employ a novel study design (considering paired transplants with differential/ dichotomous outcomes) to see what can be learned about the mechanism underlying good/ poor outcomes when donor factors, which often dominate, are controlled by design. Whilst outcome prediction is a worthy objective it is not one that we focus on in our manuscript.
Thank you for your comment regarding lack of data on immunosuppressive therapy. We agree, and have revised the manuscript to include additional data on this. As with any study making use of registry data, we are limited by the variables recorded by the registry - however we were able to show a relationship between tacrolimus use and probability of prolonged DGF, and between both tacrolimus and prednisolone use and probability of good 12-month function.
Round 2
Reviewer 1 Report
Thank you for responce.
I agree with your comment.